# Isolated Prolongation of Activated Partial Thromboplastin Time: Not Just Bleeding Risk!

**DOI:** 10.3390/medicina59061169

**Published:** 2023-06-17

**Authors:** Rita Carlotta Santoro, Angelo Claudio Molinari, Marzia Leotta, Tiziano Martini

**Affiliations:** 1Hemostasis and Thrombosis Unit, Azienda Ospedaliera Pugliese-Ciaccio, 88100 Catanzaro, Italy; ritacarlottasantoro@gmail.it (R.C.S.); marzialeotta@virgilio.it (M.L.); 2Thrombosis and Hemostasis Unit, IRCCS Istituto Giannina Gaslini, 16147 Genova, Italy; 3Immuno-Haematology and Transfusion Medicine, Center for Congenital Bleeding Disorders, Cesena General Hospital, 47521 Cesena, Italy; tiziano.martini@auslromagna.it

**Keywords:** activated partial thromboplastin time (aPTT), coagulation factor defect, mixing test, intrinsic pathway, circulating anticoagulants, bleeding patient

## Abstract

Activated partial thromboplastin time (aPTT) is a fundamental screening test for coagulation disturbances. An increased aPTT ratio is quite common in clinical practice. How the detection of prolonged activated aPTT with a normal prothrombin time is interpreted is therefore very important. In daily practice, the detection of this abnormality often leads to delayed surgery and emotional stress for patients and their families and may be associated with increased costs due to re-testing and coagulation factor assessment. An isolated, prolonged aPTT is seen in (a) patients with congenital or acquired deficiencies of specific coagulation factors, (b) patients receiving treatment with anticoagulants, mainly heparin, and (c) individuals/patients with circulating anticoagulants. We summarize here what may cause an isolated prolonged aPTT and evaluate the preanalytical interferences. The identification of the cause of an isolated prolonged aPTT is of the utmost importance in ensuring the correct diagnostic workup and therapeutic choices.

## 1. Introduction

Blood coagulation is a complex physiological process that prevents excessive bleeding and promotes wound healing. It involves a series of intricate biochemical reactions and the formation of a blood clot.

Primary hemostasis is the initial response to vascular injury. When a blood vessel is damaged, the exposed collagen fibers in the vessel wall trigger vasoconstriction, reducing blood flow to the site of injury. Simultaneously, platelets in the bloodstream become activated and adhere to the damaged endothelium at the site of injury. Platelet adhesion forms a platelet plug, temporarily sealing the damaged vessel.

Secondary hemostasis reinforces the primary platelet plug by forming a fibrin clot. It involves a cascade of enzymatic reactions which lead to the conversion of fibrinogen into fibrin. This cascade can be initiated via two main pathways: the intrinsic pathway and the extrinsic pathway.

The intrinsic pathway is activated when blood comes into contact with negatively charged surfaces, such as the collagen in the damaged vessel wall. It involves a series of clotting factors (Factors XII, XI, IX, and VIII) that sequentially activate each other, ultimately leading to the activation of Factor X.

The extrinsic pathway is initiated via the release of tissue factor from damaged tissues outside the blood vessels. The tissue factor forms a complex with Factor VII which then activates Factor X.

Activated Factor X combines with Factor V and calcium ions to form the prothrombinase complex. This complex converts prothrombin (Factor II) into thrombin (Factor IIa). Thrombin plays a central role in the coagulation cascade, converting fibrinogen into fibrin, which forms a mesh-like structure to stabilize the platelet plug and create a fibrin clot [1].

Activated partial thromboplastin time (aPTT) is a clot-based assay that is sensitive to factor defects in the intrinsic and common pathways of coagulation. The development of aPTT as a coagulation test was first reported in 1953 by Langdell [2], but thirteen years before, Dam and Venndt [3] described how diluted tissue thromboplastin, a mixture of tissue factor and phospholipids, was able to distinguish normal plasma and hemophilic plasma. Langdell et colleagues used the term “partial” to describe the thromboplastin used in this assay to distinguish it from the “complete” thromboplastin used in the prothrombin time (PT) test, which is not able to discriminate normal plasma and hemophilic plasma. In 1961, Proctor and Rapaport modified this assay by adding an activator of contact factor (“activated” PTT), kaolin, which allowed for more uniform sample activation and improved the reproducibility and predictability of the procedure’s performance, in addition to shorter clotting times and a narrower reference interval [4,5].

The aPTT is a widely used coagulation assay; for this reason, the finding of a prolongation of this clotting time in the presence of a normal PT is relatively common in several clinical settings and often represents a diagnostic challenge for clinicians and laboratory personnel.

## 2. Activated Partial Thromboplastin Time (aPTT): Rationale, Procedure, and Aims

### 2.1. What Is the Rationale of aPTT?

The activated partial thromboplastin time is sensitive to deficiencies in the activities of factors of the so called “intrinsic and common pathways”: factors II, V, VIII, IX, X, XI, XII, fibrinogen, high-molecular-weight kininogen (HMWK), and prekallikrein (PK) (Figure 1, [6]).

These aforementioned deficiencies can be:Congenital, for example, Factor VIII (hemophilia A) or Factor IX (hemophilia B);Acquired due to a neutralizing antibody (acquired hemophilia) or the effect of an anticoagulant therapy (unfractionated or low-molecular-weight heparin, LMWH, or direct oral anticoagulants, DOACs).

The type of contact factor activator, combined with the type and concentration of phospholipids in the reagent, influences the sensitivity of aPTT to the deficiencies in clotting factors, to the presence of lupus anticoagulants, and to the presence of unfractionated heparin [5].

### 2.2. How Is aPTT Performed?

At present, this assay is fully automated and is executable using modern coagulation analyzers that are capable of accurate sample dilutions, reagent additions, incubation at 37 °C, end-point clotting time measurements (optical or mechanical), and data analysis using software [5].

The reagents employed in aPTT are as follows [7]:An activator, a substance able to sustain an activation reaction activation of the zymogens belonging to the so called “contact pathway”; these are mostly factor XII but may also be HMWK and PK. Activators can be inorganic (kaolin or micronized silica) or organic (ellagic acid or vegetable phosphatides); they have different analytic sensitivities to detect factor deficiencies or the presence of inhibitors. Currently, kaolin is rarely used due to its opacity, which can inhibit the optical recognition of fibrin formation.Phospholipids, including phosphatidylserine (PS), phosphatidylcholine (PC), phosphatidylethanolamine (PE), and sphyngomyelin (SM), are incorporated into the reagent used for testing to reproduce in vitro the role of platelet in vivo.oPS serves as a surface for the assembly and activation of coagulation factors, specifically those involved in the intrinsic pathway. It provides a platform for the formation of the intrinsic tenase complex and supports the activation of Factor X, which is crucial for clot formation.oPC contributes to the overall stability and structure of the lipid vesicles used in the aPTT reagent. It helps maintain the integrity of the phospholipid membrane and aids in the presentation of other coagulation factors during the assay.oPE is involved in the formation of the phospholipid membrane used in the aPTT assay. It contributes to the overall structure and fluidity of the membrane, which are important for the proper assembly and activation of coagulation factors.oSM, like PC, is a key component of the lipid vesicles used in the aPTT reagent. It contributes to the structure and stability of the phospholipid membrane, facilitating the presentation of other necessary coagulation factors during the assay.Calcium chloride is used to reintroduce in the reaction calcium ions previously depleted by the anticoagulant (3.2% trisodium citrate) present in the blood.Citrated plasma is also used. The recommended anticoagulant for blood collection for coagulation analyses is trisodium citrate in 1 + 9 ratio with blood.

An aPTT test is performed in two steps:The addition of an activator and phospholipids to citrated plasma, determining the generation of Factors XIIa and XIa.After incubation at 37 °C, the plasma is recalcified by adding calcium chloride; beginning from this moment, the activated partial thromboplastin time is recorded as the time in seconds needed to generate the fibrin clot.

The reference interval for aPTT is typically within 20–45 s; however, this varies widely between laboratories and is dependent upon the analyzer employed, the type of activator used, etc. Therefore, the results can be better expressed as a ratio between the patient’s plasma clotting time and the clotting time of a pool of plasmas belonging to healthy subjects. The aPTT ratio is more helpful in monitoring anticoagulant therapy with unfractionated heparin and argatroban (at ratios of 2.5–3.5 and 1.5–3, respectively) and for the screening and confirmation of the presence of the lupus anticoagulant. The prolongation of a phospholipid-dependent coagulation test, the persistence of this prolongation in a mixing test with normal plasma, and the correction of this prolongation by adding an excess of phospholipids can be better demonstrated using the ratio [5,8].

### 2.3. Why Is aPTT Performed?

aPTT is a screening test (“first level”) that plays a key role in the evaluation of a patient presenting with bleeding symptoms. It is also used to monitor treatment with unfractionated heparin and argatroban (a parenteral direct thrombin inhibitor used in patients with heparin-induced thrombocytopenia and thrombosis). Finally, it is useful in screening and confirming the presence of the lupus anticoagulant [5].

## 3. Preanalytical Cause of Prolongation of aPTT and Other Coagulation Tests

Several preanalytical interferences can affect coagulation tests (Table 1).

### 3.1. Hemolysis, Hyperbilirubinemia, and Hypertriglyceridemia

Hemolysis interferes with both optical and mechanical measurement methods via optical interference due to the presence of cell-free hemoglobin absorbance and via biologic interference due to the release of molecules able to activate platelet and coagulation factors [9].

Hyperbilirubinemia and hypertriglyceridemia both interfere with optical methods. To overcome this interference, modern optical analyzers have the capability to increase the reading wavelength over 650 nm. Regardless, extreme hyperbilirubinemia and hypertriglyceridemia may make performing the coagulation test with optical analyzers unfeasible [9].

### 3.2. Blood/Anticoagulant Ratio and Hemoconcentration

As in the performance of other laboratory coagulation tests, for aPTT, an adequate amount of blood should be placed in the test tube; the optimal ratio of blood to anticoagulant should be 9:1.

The importance of this ratio is confirmed by the fact that hemoconcentration (a hematocrit higher than 55%, which could be present in patients with heart disease and in newborns, for example) prolongs aPTT as the plasma volume per blood volume is decreased, leading to an excess of anticoagulant in the sample if no correction is applied (the necessary volume of the anticoagulant is consequently decreased) [8]. The following formula has been recommended to adapt the volume of sodium citrate to the hematocrit: C = 1.85 × 10^−3^) (100-Hct) (V _Blood_), where C is the volume of citrate in the tube, Hct is the patient’s hematocrit, V is the blood amount to be collected, and 1.85 × 10^−3^ is a constant. However, a simplified method for adapting the amount of citrate to high hematocrit values has been proposed: since in most samples with high hematocrit values, the hematocrit value ranges from 55% to 65%, the removal of half the sodium citrate from the evacuate tube is sufficient to obtain reliable plasma samples [10]. On the other hand, low hematocrit values do not require the correction of the citrate volume [11].

## 4. Isolated, Prolonged aPTT: Prevalence and Causes

The causes of the prolongation of aPTT are summarized in Table 2.

### 4.1. Isolated, Prolonged aPTT: A Truly Unexpected Finding?

In an Italian retrospective study [26], 5.8% of 8.069 patients undergoing elective surgery had an abnormal aPTT; 2.9% (240 patients) had an aPTT ratio higher than 1.3 and for this reason, they were worthy of further investigation. 

An old review by Munro et al. [27] which considered 29 papers regarding the value of routine pre-operative coagulation testing showed an incidence of aPTT abnormalities of 15.6%.

A recent large Danish study [24] showed a prolonged aPTT in 12% of 18.642 aPTT measurements performed on 10.697 patients (excluding those affected by known coagulation disorders); 79% of these abnormal aPTTs were reported to be moderately or severely prolonged (40–45 or >45 s, respectively).

### 4.2. Isolated, Prolonged aPTT: What Are the Causes?

#### 4.2.1. Heparin Contamination

Heparin contamination can be a source of error and is often observed when blood is collected from a central venous line. The suspicion of this preanalytical problem should be an indication to repeat sampling from a peripheral vein [24].

However, it should be taken into account that the heparin contamination of samples destinated for use in coagulation tests could also happen if the blood is collected from a direct venipuncture that is performed after the venous lines are flushed with an amount of heparin solution containing enough heparin enough to induce an anticoagulant effect in the patient.

#### 4.2.2. C-reactive Protein

C-reactive protein (CRP) interference with the measurement of aPTT is most likely phospholipid-dependent and depends on both the CRP concentration and aPTT assay type [12,13].

#### 4.2.3. Lupus Anticoagulants

Lupus anticoagulants, a heterogeneous group of immunoglobulins that can bind β_2_-glycoprotein I, prothrombin, or other proteins in a complex with negatively charged phospholipids, are able to prolong phospholipid-dependent coagulation tests, including aPTT tests [18]. The presence of lupus anticoagulants can increase the risk of (venous or arterial) thrombosis. The phenomenon characterized by a prolongation of clotting time (aPTT) and accompanied by a thrombotic (not, as it could be expected, hemorrhagic) risk is defined as a biological paradox; very rarely, the presence of a lupus anticoagulant is associated with anti prothrombin antibodies that can determine hypoprothrombinemia and a consequent bleeding risk (lupus hypoprothrombinemia syndrome) [19].

Given the absence of population-based studies, the true prevalence of antiphospholipid–antibody positivity in the general population is not known. Between 20% and 30% of patients with systemic erythematosus lupus have persistent antiphospholipid antibodies [20]. Among patients without an autoimmune disease, the prevalence of this antibody positivity is 6% among women with pregnancy complications, 10% among patients with venous thrombosis, 11% among patients with myocardial infarction, and 17% among patients with stroke who are younger than 50 years of age [21]. About 1% of healthy blood donors are positive for a lupus anticoagulant [23]: for this reason, many laboratories use an aPTT assay with a phospholipid concentration and composition insensitive to lupus anticoagulants in vitro and a lupus-sensitive aPTT assay dedicated to investigating a suspicion of antiphospholipid syndrome [23].

#### 4.2.4. Drug Interferences

As mentioned previously, aPTT is used to monitor therapies using unfractionated heparin (UFH) and argatroban [28,29]: these drugs prolong aPTT. Low-molecular-weight heparins (LMWHs) are often reported not to affect aPTT, but some commercial brands can have this effect [11].

aPTT is also sensitive to direct oral anticoagulants (DOAC), which can cause an isolated prolongation of this clotting time, even if they often also prolong PT/INR, and aPTT is not recommended for monitoring DOAC therapy [25].

Anticoagulant therapy with vitamin K antagonists (VKAs) can also affect aPTT, but this type of therapy mainly prolongs PT, limiting the activation of factor VII.

#### 4.2.5. Acquired Hemophilia A (AHA) and Acquired Von Willebrand Disease (AVWD)

These acquired deficiencies of clotting factors due to the presence of autoantibodies are very rare diseases (incidences of ~1–2 people per million per year) and are often associated with several pathological conditions, such as autoimmune disorders, malignancies, cardiovascular diseases, pregnancy, and drug administration. AHA and AVWD can lead to severe bleeding events which are sometimes life-threatening and can require immediate intervention with anti-hemorrhagic therapy (a factor concentrate or a bypassing agent) and underlying disorder treatment (e.g., immunosuppression) [17].

A laboratory characteristic of AHA is the isolated prolongation of aPTT, while AVWD more frequently presents with reductions in both Von Willebrand factor antigen and activity; the hallmark of AHA is the persistence of this prolongation after a mixing test (as described above) due to the presence of an inhibitor antibody.

#### 4.2.6. Hemophilia A and B (HA, HB)

Hemophilia A and B are rare (incidences of ~1 in 5.000 and 1 in 30.000 male births, respectively) X-linked inherited clotting diseases caused by the deficiency of factors VIII and IX; affected patients present a prolonged aPTT and a clinical phenotype with several bleeding symptoms, whether spontaneous (in mild and moderate forms) or provoked by traumas or surgery [14]. Clinical suspicion is also based on family history. Patients with mild or moderate HA or HB can be identified from coagulation tests performed prior to surgery or for other reasons.

#### 4.2.7. Von Willebrand Disease (VWD)

Von Willebrand Disease is the most common inherited bleeding disorder (affecting up to 1% of the general population) and is characterized by a reduced or abolished synthesis or an altered function of Von Willebrand factor, which plays a role in both primary and secondary hemostasis. Patients with VWD can be asymptomatic or have spontaneous (in the most severe forms) or (more frequently) provoked bleedings after surgery, invasive procedures, traumas, delivery, etc. [30]. These patients can present an isolated, prolonged aPTT, depending on the level of factor VIII (as Von Willebrand factor is the carrier of factor VIII) in plasma; this phenomenon can be observed depending on the type and severity of VWD, whether type 3 (a deep deficiency or absence of Von Willebrand factor) or type 2N (altered function of von Willebrand factor with impaired binding with factor VIII). 

#### 4.2.8. Factor XI Deficiency

Factor XI deficiency has a prevalence of 1 in 1 million persons in most populations and is more prevalent in Ashkenazi Jews and French Basques. Its clinical picture is very heterogeneous, without a correlation between factor level and bleeding symptoms. Patients with a severe disorder are at a higher risk of bleeding, but some of them may remain asymptomatic, and patients with a partial deficiency may bleed after trauma or surgery. In particular, bleeding events happen when a fibrinolytic site is injured (nasal cavity or genitourinary tract) [15]. This congenital deficiency determines the prolongation of aPTT (which is sensitive to factor XI activity).

#### 4.2.9. Contact Pathway Factor Deficiency

Deficiencies in high-molecular-weight kininogen, prekallikrein, or factor XII cause a (sometimes remarkable) prolongation of aPTT but cannot provoke bleedings; these disorders are frequently incidentally diagnosed from screening before surgery and are important for differential diagnosis with the other causes of isolated prolongation of aPTT [15].

## 5. Isolated, Prolonged aPTT: Differential Diagnosis

### How can We Identify the Cause of Isolated, Prolonged aPTT?

First step—possible pre-analytical interferences?

First, evaluate the appropriateness of the sample; exclude that the sample is jaundiced, hemolysis, or hyperlipemic. Verify the correct blood/anticoagulant ratio, namely, the adequate sample filling and hematocrit [8,9,10].

Second step—presence of anticoagulants?

Then, the presence of interfering drugs (heparin and DOAC) should be excluded; performing thrombin time and anti-Xa assays will provide information about the type of drug: Heparin will be detected by both tests;Direct anti-IIa inhibitor will be detected only via thrombin time;Direct anti-Xa inhibitors will be detected only by an anti-Xa assay.

Reptilase time is a functional test based upon the enzymatic activity of a snake venom, Batroxobin, that is able to cleave fibrinogen in a different site compared to thrombin. For this reason, this clotting time is not sensitive to heparin, direct anti-IIa inhibitors, or direct anti-Xa inhibitors. As these drugs are able to affect thrombin time, comparing reptilase time and thrombin time allows for these situations to be discriminated [5,25,31].

Third step—inhibitor or factor deficiency?

In presence of prolonged aPTT, after the exclusion of plasma contamination by anticoagulant drugs, a mixing test should be performed to discriminate whether the prolongation of aPTT is due to the deficiency of one or more coagulation factors (factor XII, prekallikrein, high-molecular-weight kininogen, factor XI, factor IX, or factor VIII) or to the presence of a circulating anticoagulant directed against one or more of the same factors or against membrane phospholipids [7].

The rationale of the test is that if the cause of the aPTT prolongation is a deficiency, the aPTT performed after mixing the patient’s plasma with a “normal” plasma (whose characteristics are described above) will be normalized [16]. On the other hand, the presence of a circulating anticoagulant will not induce the normalization of aPTT performed on the mixture because this antibody will inhibit both the patient’s factor(s) and the normal plasma’s factor(s).

The mixing test can be performed by setting up the mixing manually or in an automated way, employing the same reagents used for aPTT with the addition of a “normal” plasma, which is a pool of plasmas (normal pool plasma, NPP) prepared using a mixture of plasmas obtained from at least twenty healthy adult subjects (preferably in an equal proportion of males/females) with negative personal and familiar histories for coagulopathies and documented absences of lupic anticoagulants [5,32]. NPP can be prepared *“in house”* by packing small fractions of mixed plasmas that can be frozen and stored (−70 °C) for at least one year. Immediately prior to use, the fractions will be quickly defrosted and used within two hours [7]. Lyophilized commercial NPPs presenting the same characteristics can also be used.

NPP is assumed to contain activity levels of all the coagulation factors between 50 and 200 U/dL, with an average value near 100 U/dL: this fact allows the mixing procedure to obtain an activity level of the deficient factor(s) in the test mixture that is near to the value needed to generate an APTT that falls in the normal range (namely~50 U/dL), even in the extreme (rare) condition of totally deficient patient plasma [6].

Before performing a mixing test, it is necessary to remember that some circulating anticoagulants, such as the autoantibodies directed against FVIII in acquired hemophilia A, require incubation (two hours at 37 °C) to fully explain their activity. In this case, the inhibitor effect is time- and temperature-dependent, namely, its action is not immediately evident but needs specific conditions of time and temperature (that must be reproduced in the laboratory), and it is called type 2 kinetic. Nevertheless, even if time-dependence is a general rule, there are some exceptions (for example, if present at a high titer, autoantibodies directed against FVIII can have an immediate effect on the mixing) [5,32,33].

Taking these aspects into consideration, in different situations, based on appropriate clinical–anamnestic considerations, the aPTT assay can be performed on the test mixture immediately or after incubation:For an immediate mixing test, the plasma is prepared using equal volumes (1:1) of NPP and patient plasma, and aPTT is performed on it at room temperature [7]. In parallel, as a control, aPTT is performed on the NPP. Chang et al. [34] suggested that the aPTT correction in a mixing test with a 4:1 ratio of NPP and patient plasma can achieve better sensitivity and specificity, mostly in the situations in which the antibody power is relatively weak. Recently, a Chinese study [35] on 251 samples demonstrated that the aPTT mixing studies had good sensitivity and specificity in differentiating factor deficiencies from inhibitors (and in differentiating time-dependent from time-independent inhibitors) and that the combination of 1:1 and 4:1 mixing studies was able to improve the diagnostic ability compared with the 1:1 ratio alone in those cases characterized by a borderline Rosner index value (see below);An incubated mixing test is realized by incubating the mixture at 37 °C for 120 min before performing aPTT (and also incubating the NPP and patient plasma separately before performing aPTT as a control) [5,33].

In an optimal scenario, performing in sequence a mixing study at room temperature and a mixing study with 2 h of incubation (even if the normalization of aPTT was already achieved without incubation) could be an ideal approach to excluding the presence of different thermal sensitivities; nevertheless, in real practice, most laboratories rarely routinely perform mixing tests with incubation, leading to missed diagnoses [16]. 

One of the critical points of the mixing test is the interpretation of the results. Two different methods have been proposed:The normality range method, in which the aPTT value measured from the mixture must fall within the normal range (a range determined by the laboratory that is effective for the specific combination of the reagent and coagulation analyzer used for the measurement). The advantages of this method are its easiness and immediacy; it has an important diagnostic power in the situations in which the patient plasma aPTT is markedly prolonged. However, it shows its weakness in situations in which aPTT is only mildly prolonged (because the mixing could determine an aPTT value in the normal range if there are both a factor deficiency and a low titer inhibitor due to inhibitor dilution) [36,37];The index of circulating anticoagulant (ICA) method, ICA (or Rosner index), is defined by the following formula:
ICA = [(CT_mixing_ − CT_NPP_)/CT_patient plasma_] × 100

The higher the ICA value, the higher the probability that the mixing will not correct aPTT. The cut-off value is usually fixed at 10–15% [5,32,33,38,39], but it depends on the combination of the reagent/analyzer (which should be determined in every single laboratory). ICA has been demonstrated to be helpful in predicting the presence of lupus anticoagulants with great accuracy [39].

The combination of the two mixing procedures (immediately tested and incubated at 37° for 120 min) offer the possibility to differentiate between a factor deficiency, a lupus anticoagulant, and a specific inhibitor (Table 3).

Fourth step—suspicion of factor deficiency:

When the mixing test suggests the presence of a factor deficiency, specific factor assays are needed to determine which factor is deficient: the factors whose deficiencies are univocally related to an isolated prolongation of aPTT are factor VIII, factor IX, factor XI, and factor XII [7]. The other components of the contact pathway (high-molecular-weight kininogen and prekallikrein) are not routinely tested in all laboratories because of their lack of association with a bleeding phenotype. The prolongation of the incubation of the aPTT testing mixture of up to 1 h at 37 °C is useful for differentiating prekallikrein and kininogen defects. Prekallikrein (Fletcher factor) deficiency could be suspected if every long aPTT in an asymptomatic patient is corrected by prolonging the incubation, as Factor XII autoactivates via incubation in the absence of prekallikrein [37]. On the other hand, in the presence of normal levels of Factor XII in an asymptomatic patient whose aPTT does not autocorrect after prolonged plasma incubations, the suspected diagnosis should be a high-molecular-weight kininogen defect [40].

If a factor VIII deficiency is found, it is recommended to test also for Von Willebrand factor (activity and antigen) [27].

Fifth step—suspicion of factor inhibitor:

To discriminate the type of inhibitor, its necessary to take the following actions [24]:Test for lupus anticoagulants by performing the dilute Russell viper venom time (dRVVT): the presence of a lupus anticoagulant can be diagnosed by finding a ratio between a phospholipid-rich dRVVT assay (confirm) and a phospholipid-poor assay (screen) ≥1.3. A negative dRVVT test does not allow for the exclusion of lupus anticoagulants as certain antibodies of this class are only detected using an aPTT-derived assay.Perform specific factor assays (VIII, IX, XI, and XII), which allow for the identification of the specific deficient factor. The next step is the titration of the inhibitor (typically of the inhibitor against factor VIII in acquired hemophilia A); in a Bethesda assay, the presence of lupus anticoagulants will be able to interfere with the test [41].

Lupus anticoagulants, mainly the ones with high titers, can also interfere with the measurement of the intrinsic pathway factors (VIII, IX, XI, and XII), whose values could all appear decreased if measured with one-stage assays with lupus-sensitive reagents [42,43]. It is important to be aware of this phenomenon to avoid the misinterpretation of a lupus anticoagulant as a specific factor inhibitor: if Factor VIII or Factor IX appear reduced, the measurement of these factors’ activities using a chromogenic method will exclude (or confirm) the presence of a deficiency. The interference of LA in the one-stage assay of FXI and FXII could be depicted by performing the assay after further dilutions of the sample. Occasionally, there may be the presence of both lupus anticoagulants and a single factor inhibitor, for example, together with a common pathway factor inhibitor (such as a factor V inhibitor); in this case, it can be useful to perform the mixing studies for both PT/INR and aPTT as the presence of a common pathway inhibitor can be excluded if the mixing corrects PT/INR but not aPTT [44].

## 6. Conclusions

After about seventy years from its first description in medical journals, aPTT is still a milestone in coagulation screening. The modern automated procedures offer prompt and reliable results; in combination with thrombin time and the mixture test, it is essential in the diagnostic workup of many alterations of the intrinsic coagulation pathway. However, the isolated prolongation of aPTT is commonly found since as in other clotting tests, aPTT is prone to pharmacological interferences and preanalytical disturbances that must be well known in order to make the correct decisions in the clinic and in the laboratory.

## Figures and Tables

**Figure 1 medicina-59-01169-f001:**
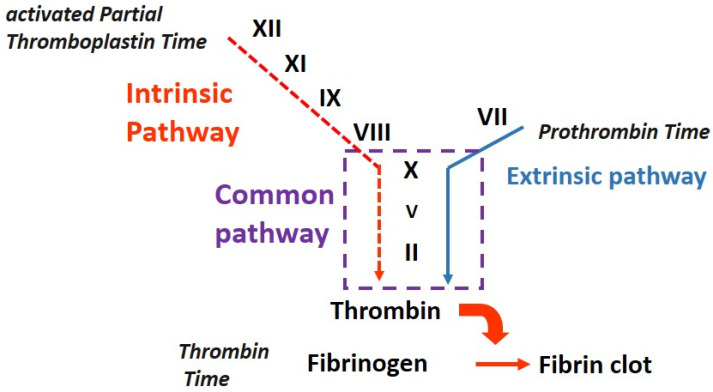
Sequence of events that occur during secondary hemostasis and the role of screening tests [1]. The intrinsic pathway is initiated when blood comes into contact with negatively charged surfaces, such as collagen exposed in the damaged vessel wall, activating Factor XII. Factor XIIa activates Factor XI to its active form, Factor XIa. Factor XIa, along with its cofactor, Factor VIII, activates Factor IX to Factor IXa. Factor IXa, in the presence of Factor VIII and calcium ions, activates Factor X to Factor Xa. The extrinsic pathway is initiated via the release of tissue factor from damaged tissues outside the blood vessels. Tissue factor forms a complex with Factor VII, leading to the activation of Factor VII to its active form, Factor VIIa. Factor VIIa, along with tissue factor, activates Factor X to Factor Xa. Both the intrinsic and extrinsic pathways converge at Factor Xa, which is a key enzyme in the coagulation cascade. Factor Xa combines with Factor V and calcium ions to form the prothrombinase complex, which converts prothrombin (Factor II) into thrombin (Factor IIa).Thrombin plays a central role in the coagulation cascade, converting fibrinogen into fibrin. The two common screening tests, prothrombin time (PT) and activated partial thromboplastin time (aPTT), help assess the clotting ability of the blood and detect any abnormalities or deficiencies in the clotting factors. Prothrombin time measures the extrinsic pathway of the coagulation cascade. It assesses the time taken for the formation of a clot after the addition of tissue factor to plasma and primarily evaluates the function of factors I, II, V, VII, and X. Activated partial thromboplastin time (aPTT) measures the intrinsic pathway of the coagulation cascade. It evaluates the time taken for the formation of a clot after the addition of an activator to plasma and primarily assesses the function of factors I, II, V, VIII, IX, X, XI, and XII.

**Table 1 medicina-59-01169-t001:** Preanalytical causes of prolonged coagulation tests.

Cause
Hemolysis
Hyperbilirubinemia
Hypertriglyceridemia
Blood/anticoagulant ratio
Hemoconcentration

**Table 2 medicina-59-01169-t002:** Causes of isolated and prolonged aPTT.

Clinical Condition	Ancillary Laboratory Data (Refer to Section 2.2)	Note	Reference
High C-reactive protein	Normal prothrombin time	Variable sensitivity among aPTT reagents	[12,13]
Factor VIII deficiency (hemophilia A)	Normal thrombin time Corrected immediate mixing test	Possible positive bleeding history	[14]
Factor IX deficiency (hemophilia B)	Normal thrombin time Corrected immediate mixing test	Possible positive bleeding history	[14]
Factor XI deficiency	Normal thrombin time Corrected immediate mixing test	Possible positive bleeding history	[15]
Factor XII deficiency	Normal thrombin time Corrected immediate mixing test	No bleeds	[15]
Pre kallicreine deficiency	Normal thrombin time Corrected both immediate and incubated mixing tests, as well as aPTT incubated up to 60 min	No bleeds	[16]
Factor XII and high molecular kininogens deficiency	Normal thrombin time Corrected immediate mixing test	No bleeds	[16]
FVIII inhibitor	Normal thrombin time Immediate mixing test may be corrected Incubated mixing test usually uncorrected	Possible positive bleeding history	[17]
Lupus anticoagulant (LA)	Normal thrombin time Uncorrected mixing test, immediate and incubated	See text	[18,19,20,21,22,23]
Unfractionated heparin	Uncorrected mixing test Prolonged thrombin time	Positive history for the drug	[24]
Low molecular weight heparin/fondaparinux	aPTT prolonged only at therapeutic doses Presence of antiXa activity	Positive history for the drug	[25]

**Table 3 medicina-59-01169-t003:** Utility of incubation in PTT mixing procedure [39].

Immediate Correction	Correction after Incubation	Conclusion
Yes	Yes	Factor deficiency
No or Partial	No	Factor inhibitor
No	No	Lupus anticoagulant

## Data Availability

Not applicable.

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
