# Peer review of "Isolated Prolongation of Activated Partial Thromboplastin Time: Not Just Bleeding Risk!"

_medicina, 2023, doi:10.3390/medicina59061169_

Round 1

Reviewer 1 Report

The knowledge summarized in the article is an important topic for clinicians and laboratory doctors.

However, some corrections should be made.

1.In the abstract APTT and aPTT are used. The abbreviations should be unified.

2. In the Section 2.1 Figure 1 is mentioned. The picture is taken from the [5] source. Has permission been obtained to use the figure?

3. Rows 64-65: The aim of the manuscript is „  The identification of the cause of an isolated prolonged APTT“, so, the sentence„aPTT is also sensitive to lupus anticoagulant (LA), because of the affinity of these kinds of antibodies for the phospholipids used in the assay” should be removed. I would suggest not write about disorders when PT also changes together with aPTT in the manuscript.

4. 59 row: it is not clear, what exactly authors mean using the phrase „These deficiencies...“.  It should be specified.

5. According to Section 4 , the title of the Table 2 should be „Causes of isolated prolonged aPTT“. I suggest to give only isolated aPTT causes in the table and supement it with knowledge about laboratory reasons if it is possible (because the methodology is given in 2.2 section). Referencies used for table formation should be given in the additional column of the table.

6. 5.1 section rows 274-291 has not referencies.  Do these statements are made by the authors‘ experimental work ? If not, the referencies have to be added.

7. Referencies used for Table 3 formation should be added after the Title of table 4.

8. Rows 378-390 are without the referencies too. Are the given information gained by the authors? If not, the referencies should be added.

Author Response

Response to Reviewer 1 Comments

Point: .In the abstract APTT and aPTT are used. The abbreviations should be unified.

Response 1: We unified the abbreviation as requested

Point 2:. In the Section 2.1 Figure 1 is mentioned. The picture is taken from the [5] source. Has permission been obtained to use the figure?

Response 2: We thanks the reviewer for this question. We created a new picture.

Point 3: . Rows 64-65: The aim of the manuscript is „ The identification of the cause of an isolated prolonged APTT“, so, the sentence„aPTT is also sensitive to lupus anticoagulant (LA), because of the affinity of these kinds of antibodies for the phospholipids used in the assay” should be removed. I would suggest not write about disorders when PT also changes together with aPTT in the manuscript.

Response 3: We thanks the reviewer for this advice. We accordingly removed the sentence “aPTT is also sensitive to lupus anticoagulant (LA), because of the affinity of these kinds of antibodies for the phospholipids used in the assay” – now rows 100-104.

Point 4: 59 row: it is not clear, what exactly authors mean using the phrase „These deficiencies...“. It should be specified

Response 4: We agree that the sentence was no clear. We modified it (row 96).

Point 5: According to Section 4 , the title of the Table 2 should be „Causes of isolated prolonged aPTT“. I suggest to give only isolated aPTT causes in the table and supplement it with knowledge about laboratory reasons if it is possible (because the methodology is given in 2.2 section).
References used for table formation should be given in the additional column of the table

Response 5: We appreciate this advice from the reviewer. We changed the title accordingly and added the column with the references, as already suggested also from reviewer Nr 2.

Point 6: 5.1 section rows 274-291 has not references. Do these statements are made by the authors‘ experimental work ? If not, the references have to be added

Response 6: We added the references to the statements. (now rows 326-342)

Point 7. References used for Table 3 formation should be added after the Title of table 4

Response 6: We added the reference after the title.

Point 8: Rows 378-390 are without the references too. Are the given information gained by the authors? If not, the references should be added.

Response 8: We added the due references. (now rows 426-440)

Reviewer 2 Report

In the present review, Santoro et al. demonstrated the importance of activated partial thromboplastin time (APTT) test to assess coagulation disorders which significantly plays a part in clinical practice, e.g., bleeding complications often leading to unnecessary delay during surgery further contributing to stress for the patients. The manuscript is well-organized, well-written and of substantial importance in the field of bleeding disorders. However, I have a few minor suggestions which could improve the present form of the manuscript as follows:

1.       In the introduction, the authors should start with the definition of blood coagulation, a very brief description of intrinsic and extrinsic pathways and defects in coagulation factors could lead to hypo- and hypercoagulable conditions with appropriate references which should be followed by the aPTT description.

2.       A brief description of classical coagulation cascade model and the role of screening tests is required in the legend of Figure 1.

3.       Page 2, line 60 congenital hemophilia A or B- mention at the first place which factor is responsible for that FVIII or FIX, respectively.

4.       In section 2.2, a brief discussion is needed how different phospholipids (such as phosphatidylserine, phosphatidylcholine, phosphatidylethanolamine, sphingomyelin etc.) influence the coagulation process.

5.       One panel in Table 2 is needed for references.

6.       In section 4.2.5, the authors should address how recombinant FVIIa is used as a bypassing agent to treat hemophilia and cite the recently published important references [Factor VIIa induces extracellular vesicles from the endothelium: a potential mechanism for its hemostatic effect. Blood. 2021 Jun 17;137(24):3428-3442. doi: 10.1182/blood.2020008417; Factor VIIa treatment increases circulating extracellular vesicles in hemophilia patients: Implications for the therapeutic hemostatic effect of FVIIa. J Thromb Haemost. 2022 Aug;20(8):1928-1933. doi: 10.1111/jth.15768].

7.       In section 4.2.7, the authors should briefly mention treatment measures of VWD.

A minor editing of English language is required for the better understanding of the manuscript.

Author Response

Response to Reviewer 2 Comments

Point 1: In the introduction, the authors should start with the definition of blood coagulation, a very brief description of intrinsic and extrinsic pathways and defects in coagulation factors could lead to hypo- and hypercoagulable conditions with appropriate references which should be followed by the
aPTT description.

Response 1: we appreciate the advice of the reviewer; we inserted a short description of coagulation reactions as well as of intrinsic and extrinsic patways accordingly. However, we did not add description of how defects in coagulation factors could lead to hypo- and hypercoagulable conditions as this topic is mentioned in section 2.1

Point 2: A brief description of classical coagulation cascade model and the role of screening tests is required in the legend of Figure 1

Response 2: We added a brief description of classical coagulation cascade model and the role of screening tests into the legend of figure 1 (text box format, italicized); to avoid formatting problems we separated the figure and the legend in two different pages

Point 3: Page 2, line 60 congenital hemophilia A or B- mention at the first place which factor is responsible for that FVIII or FIX, respectively

Response 3:  We modified the text accordingly (now row 96)

Point 4: In section 2.2, a brief discussion is needed how different phospholipids (such as phosphatidylserine, phosphatidylcholine, phosphatidylethanolamine, sphingomyelin etc.) influence the coagulation process.

Response 4:  we elaborated on the role of phosphatidylserine, phosphatidylcholine, phosphatidylethanolamine and sphingomyelin in the the coagulation process.

Point 5: 5. One panel in Table 2 is needed for references

Response 5: we added a column to the table where we inserted references

Point 6: In section 4.2.5, the authors should address how recombinant FVIIa is used as a bypassing agent to treat hemophilia and cite the recently published important references [Factor VIIa induces extracellular vesicles from the endothelium: a potential mechanism for its hemostatic effect. Blood. 2021 Jun 17;137(24):3428-3442. doi: 10.1182/blood.2020008417; Factor VIIa treatment increases
circulating extracellular vesicles in hemophilia patients: Implications for the therapeutic hemostatic effect of FVIIa. JThromb Haemost. 2022 Aug;20(8):1928-1933. doi: 10.1111/jth.15768]

Point 7: In section 4.2.7, the authors should briefly mention treatment measures of VWD.

Response to points 6 and 7:  We apologize with reviewer 2 but we do not intend to modify the manuscript in accordance with these two suggestions from him, because our work aims to help clinicians navigate when they obtain a prolonged aptt. It is not within the scope of this paper to discuss therapeutic choices, which would require at least one more manuscript. We are confident that the reviewer will agree with us.